# A Compact Broadside Coupled Stripline 2-D Beamforming Network and Its Application to a 2-D Beam Scanning Array Antenna Using Panasonic Megtron 6 Substrate

**DOI:** 10.3390/s24020714

**Published:** 2024-01-22

**Authors:** Jean Temga, Takashi Shiba, Noriharu Suematsu

**Affiliations:** 1Millimeter Wave Technologies, Intelligent Wireless System, Silicon Austria Labs (SAL), 4040 Linz, Austria; 2Research Institute of Electrical Communication, Tohoku University, Miyagi 980-0812, Japan; takashi.shiba.e2@tohoku.ac.jp (T.S.); noriharu.suematsu.a3@tohoku.ac.jp (N.S.)

**Keywords:** broadside coupled stripline (BCS) coupler, fifth generation (5G), two-dimension beamforming network (2-D BFN), panasonic megtron 6 substrate

## Abstract

This article presents a 4-way 2-D butler matrix (BM)-based beamforming network (BFN) using a multilayer substrate broadside coupled stripline (BCS). To achieve the characteristics of a compact, wide-bandwidth, high-gain phased array, a BCS coupler is implemented using the Megtron 6 substrate. The compact 2-D BFN is formed by combining planarly two horizontal BCS couplers and two vertical BCS couplers. The BFN is proposed without a crossover and without a phase shifter, generating phase responses of ±90° in the *x*- and *y*-directions, respectively. The proposed BFN exhibits a wide operating band of 66.7% (3–7 GHz) and a compact physical area of just 0.25 λ_0_ × 0.25 λ_0_ × 0.04 λ_0_. The planar 2-D BFN is easily integrated with the patch antenna radiation elements to construct a 2-D multibeam array antenna that generates four fixed beams, one in each quadrant, at an elevation angle of 30° from the broadside to the array axis when the element separation is 0.6 λ_0_. The physical area of the 2-D multibeam array antenna is just 0.8 λ_0_ × 0.8 λ_0_ × 0.04 λ_0_. The prototypes of the BCS coupler, the 2-D BFN, and the 2-D multibeam array antenna were fabricated and measured. The measured and simulated results were in good agreement. A gain of 9.1 to 9.9 dBi was measured.

## 1. Introduction

Due to the 5th Generation (5G) of mobile communication’s primary goals of achieving higher rate data transfer, low latency, compact devices, higher spectral efficiency, large channel capacity with wide scanning coverage, and higher power efficiency, wireless communications have seen tremendous growth in recent years [1]. By boosting the signal-to-noise ratio (SNR) at the receiver without increasing the transmission power, beam-steering antennas can greatly increase the reliability and performance of communication networks [2,3,4]. A beamforming network (BFN), such as the butler matrix [5,6], reflector lens [7,8], and Rotman lens [9,10], is highly common for creating a desired multibeam array in 1-D and 2-D, depending on various practical scenes. Due to its low cost, simpler structure, identical current paths, and improved port isolation that can be integrated into a small number of components, the BM exhibits more promising development prospects when it comes to creating a multibeam array antenna [11].

Early reports in the open literature were primarily concerned with 1-D and 2-D multibeam antennas using various guiding technologies, including microstrip line, substrate-integrated waveguide (SIW), grounded co-planar waveguide (GCPW), printed gap waveguide (PRGW), and rectangular coaxial. Only one dimension can be used to steer the 1-D radiation pattern.

The 2-D multibeam antennas have gained increased attention from researchers recently because they offer beams in two orthogonal planes, improving their versatility in spatial coverage [12,13]. However, compared to 1-D, the beamforming network (BFN) for 2-D multibeam antennas is significantly more involved. The traditional method for creating a 2-D BFN involves directly combining two sets of sub-BFNs that are spatially orthogonal [11,12,13,14,15,16], typically characterized by a spatial framework and large size. Many studies have concentrated on the planar design of 2-D BFNs as a means of resolving this problem [11,17,18], typically at the cost of an increased size. A 2-D BFN was created and designed as a planar configuration in [19]. With the use of substrate-integrated waveguide (SIW) technology, another planar 2-D BM is made possible [13]. The designs in [13,19] were, however, implemented at the expense of size expansion. Therefore, the circuit’s shrinking is essential. The size of the BFN should be less than or equal to that of the antenna array in order to produce a compact beamforming array antenna. Such circuits benefit from being compact, lightweight, easy to integrate with array antennas, more flexible, and more durable.

In this study, a 2-D multibeam array antenna is realized using the broadside coupled stripline (BCS) multilayer technique. References [20,21,22,23,24,25,26] contain multiple PCB designs of compact 1-D and 2-D BFNs employing a BCS that were proven by the reference authors. The use of the crossover and 45° phase shifter is required in the typical 1-D BFN [22], which leads to geometric complexity and performance deterioration. Two 3 dB/90° cascaded broadside couplers can be used to perform the crossover; however, doing so will result in higher losses and a larger overall system. Without using a crossover, the BFN in [23] was revised in [23,25,26]. It appears that the bandwidth is considered to be limited by the phase shifter, making design more difficult. Therefore, the implementation of a 1-D BFN topology without a crossover and phase shifter is seen as having a substantial advantage in terms of complexity, size, BW, losses, and phase errors [11,27].

The BM topologies presented in [11,27,28] served as the basis for the main concept of the BFN without a crossover and phase shifter. The BM topologies are identical with the exception of the guiding technology, which is the BCS in this paper’s design. The guiding method employed in [29] was the printed ridge gap (PRGW), whereas [27,28] used the traditional microstrip line and SIW, respectively. However, they are heavy, expensive, and exposed to erroneous radiation. In line with that, the authors of this research employ the same BM topology with the BCS wave-guiding technology, which has the properties of compact and broadband circuits. Conventional branch line couplers, as previously indicated, restrict BW, increase losses, and enlarge circuits [29].

The main advantage of a BCS is its configuration, which allows the realization of practical planar and compact circuits with a high coupling factor and wide bandwidth [21]. This improves network isolation by minimizing the interference from the antenna radiation into the BFN layer. A 2-D BFN without a crossover and phase shifter in the BCS was designed and fabricated using a Rogers substrate [24] by the same authors. To achieve the goal of a compact planarized 2-D BFN, a broadside coupler was designed in Figure 1a, and a set of two vertical couplers and a set of two horizontal couplers were connected to realize a 2-D BFN, as shown in Figure 1b.

The results obtained using the Rogers substrate RO440F showed low performance [24]. This may be due to the electrical and transmission characteristics of the Rogers substrate at lower frequencies and the adverse effects of the fabrication of circuits with a high layer count and multiple lamination cycles using the Rogers substrate [30]. To improve the performance of the fabrication with multiple laminate cycles, layers, and via holes to connect the layers, the Panasonic Megtron 6 [30] was chosen for this fabrication. It offers many advantageous features, especially for PCB designs. Low transmission loss, thick layer, strong thermal resistance, high-density interconnect (HDI), and improved stability are some of the advantages. It can even stop signal reflections and is compatible with lead-free soldering. References [31,32] present examples of circuit design and fabrication utilizing the Panasonic Megtron 6.

The rest of this paper is structured as follows: The substrate configuration and vertical via hole operation method are presented in Section 2. The design and construction of the feeding network employing the Panasonic Megtron 6 substrate, including the broadside coupler, are also shown in Section 2. The multibeam antenna array’s simulated and measured results, including the antenna element and manufactured substrates, are shown in Section 3. The paper is concluded in Section 4.

## 2. 2-D Beamforming Network

A BCS demonstrates the benefits of non-dispersion, strong anti-interference, downsizing, and broadband circuits as a planar structure. Figure 2 displays its cross-sectional schematic. It has two striplines and two ground planes. Two different types of Megtron 6 substrates, R5775N (ε_r_ = 3.37, *tanδ* = 0.002) and R5775NJ (ε_r_ = 3.34, *tanδ* = 0.003), are used to separate the metal layers. Figure 3 presents the suggested 2-D BFN. In Figure 4b, the proposed BFN is depicted. While the BFN is designed at the connected stripline layers, the patch antenna array is designed at the higher metal layer. Green designates the upper stripline conductor, whereas red designates the lower stripline conductor.

The suggested BFN architecture lacks a phase shifter and a crossover, as seen in Figure 3. There are only four broadside couplers in it. The BFN size is decreased to 0.25 λ_0_ × 0.25 λ_0_ as a result. The output ports P5, P6, P7, and P8 produce the phase difference (α_x_, α_y_) of the adjacent output ports when the input ports P1, P2, P3, and P4 are excited, where x and y are the phase shifts in the *x*- and *y*-directions, respectively. Table 1 lists the theoretical values of α_x_ and α_y_ for the associated stimulated ports.

### 2.1. Design of a Broadside Coupler

The BCS structure is given an additional layer to design the coupler, as shown in Figure 4a. A 50 Ω transmission line (TML) that is built at the upper metal layer feeds the coupler. The layers were connected by three different types of vias. The two striplines were connected using pcvia1. The BFN is connected to the antenna layer using pcvia2, while the two ground planes are interconnected using pcvia3 to balance the potentials and get rid of the capacitance effect. The metallization is 38 µm. As shown in Figure 1, a drill was used to make fabrication easier on the substrate’s backside. The striplines’ width and optimum vias’ diameter are both 0.4 mm.

The broadside coupler layout is illustrated in Figure 4b. It is composed of two quarter-waves of coupled striplines (pc3 and pc2). The striplines are close enough in proximity (0.13 mm) so that energy from one stripline passes to the other stripline. In the 3.5 GHz to 6.5 GHz frequency range, the reflection coefficient and isolation illustrated in Figure 5 are better than −10 dB. At output ports P2 and P3, an equal power division of −3 ± 0.5 dB can be seen. Over the frequency spectrum of interest, a ±90° ± 5° phase difference between the coupled and direct ports is attained. Measurements were made on the manufactured prototype depicted in Figure 4c. The measured S-parameters and the measured output port phase difference are illustrated in Figure 6. The measured and simulated results are in good agreement.

### 2.2. Design of a 2-D BFN

The 2-D BFN simulation setup is shown in Figure 7a. The 2-D BFN is extended along the same stripline length in order to connect it to the TML. The inputs and outputs of the BFN are then connected to a 50 Ω transmission line (TML) of the same length that is built into the top metal layer with a via (pcvia2). Figure 8a–d show the simulated reflection coefficients, transmission coefficients at port 1, and output port phase difference in the *x*- and *y*-directions, respectively. Over the frequency range of 3 GHz to 6.5 GHz, the simulated reflection coefficients in Figure 8a are better than −10 dB. The amplitude distribution at Port P1 is depicted in Figure 8b.

The manufactured prototype seen in Figure 7b was measured, and the outcomes are displayed in Figure 9. In the frequency range of 3–6.5 GHz, the reflection coefficients are better than −10 dB. In Figure 9b, a maximum amplitude variation of 1 dB is seen at 5.2 GHz. In Figure 9c,d, the phase dispersion in the *x*- and *y*-directions displayed a peak-to-peak error of 8° in the frequency range of 3–7 GHz. In the same frequency range of interest, the measurement performs similarly to the simulation.

The output ports produce an average equal power division of −6.25 dB when input port 1 is excited, which is almost the same as the theoretical value (−6 dB). A phase difference of −90° ± 5° along the *x*-axis is obtained at the output ports when input ports P1 and P2 are excited. In the same *x*-direction, a phase shift of +90° ± 5° is produced at the output ports when P3 and P4 are excited. In the *y*-direction, a phase difference of −90° ± 5° is produced when ports P1 and P3 are excited. A +90° ± 5° phase difference is obtained when ports P2 and P4 are excited.

## 3. 2-D Scanning Array Antenna

To evaluate the scanning performance of the proposed 2-D BFN, a 2 × 2 patch array antenna is integrated into the 2-D BFN using the vertical via transition pcvia2. A microstrip patch antenna designed at the top metal layer was chosen to simplify the architecture and the design.

### 3.1. 2 × 2 Patch Array Antenna Configuration

The design dimensions and configuration of the 2-D array antenna in a 3-D EM simulator are shown in Figure 10. The 50 Ω TML utilized to feed the array antenna is linked to the input ports of the 2-D BFN, while the output ports are connected to the antenna feeding points. To maintain the same phase difference, the striplines are all the same length. Each antenna input port has pads that make it easy to fix the SMAR006D00 End Launch connectors for measurement. The array’s size is 0.8 λ_0_ × 0.8 λ_0_. Figure 10 depicts the suggested patch-antenna element’s structure and dimensions. Figure 11 displays the simulation and measurement outcomes for the antenna element’s reflection coefficient and radiation pattern. At 5.2 GHz, the simulated and measured gains are 5.92 dBi and 5.89 dBi, respectively.

The different boundary conditions between the simulation and measurement are what cause the discrepancy in the radiation patterns at ±90° between the two. The measurement boundary condition does not take into account the backside of the DUT when it is mounted horizontally on the rotating table, but the simulation boundary condition takes into account the complete empty area around the antenna. According to Figure 11b, the antenna element’s simulated cross-polarization is less than −30 dB.

### 3.2. Calculated and Simulated Beams

The phase shift and inter-element distance for a planar rectangular array adhere to the requirements [33] specified by Equations (1)–(4).
(1)$$\alpha_{x}=-k_{0}d_{x}sin\theta_{0}cos\phi_{0}$$
(2)$$\alpha_{y}=-k_{0}d_{y}sin\theta_{0}sin\phi_{0}$$
where $\alpha_{x}$ and $\alpha_{y}$ are the phase differences in the *x*- and *y*-directions, respectively, and $k_{0}$ is the free-space wave number. The interelement distances in the *x*- and *y*-directions are 0.6 λ_0_. Equations (3) and (4) are used to express the main beam scanning direction ${(\theta }_{0},\phi_{0})$.
(3)$$\theta_{0}={sin}^{-1}\sqrt{{\left(\frac{\alpha_{x}}{k_{0}d_{x}}\right)}^{2}+{\left(\frac{\alpha_{y}}{k_{0}d_{y}}\right)}^{2}},$$
(4)$$\phi_{0}={tan}^{-1}\left[\frac{\alpha_{x}d_{x}}{\alpha_{y}d_{y}}\right],$$

The simulated 3-D radiation pattern of the four generated beams at the 5.2 GHz frequency band that corresponds to each excited input port is shown in Figure 12. The port number, beam direction, and realized gain are used to mark each beam. Geometrically speaking, coupled ports (P1, P4) and (P2, P3) are symmetric. The range of the simulated realized gain is 10.1 dBi to 10.6 dBi. Equations (1)–(4) are used to determine the theoretical beam directions using the inter-element distance *d*_x_ = *d*_y_ = 0.6 λ_0_ (34.55 mm), and the theoretical phase shifts are listed in Table 1. When ports #1 to #4 are activated, the computed maximum beam aiming angles are (30°, 45°), (30°, 135°), (30°, 225°), and (30°, 315°) in spherical coordinates, respectively. Table 2 contains a comparison of the calculated findings with the simulated results. The calculated and simulated outcomes have a respectable level of agreement. It is likely that the imbalances of the simulated phase and amplitude are what cause the disparity between the calculated and simulated beam directions.

### 3.3. Fabricated Prototype and Measurement

The fabricated prototype of the 2 × 2 array antenna is depicted in Figure 13. As stated in the introduction, the metallization, the optimal via diameters, the stripline width, and the 50 Ω TML width are each 0.038 mm, 0.4 mm, 0.4 mm, and 1.4 mm, respectively. The laminate layers are firmly pressed during the fabrication process to provide good interlayer transmission. As shown in Figure 1, a drill was used to make it easier to fabricate each feeding through the via (pcvia2) on the substrate’s backside.

### 3.4. Scattering Parameters

The measurements of S-parameters were carried out using the KEYSIGHT FieldFox Microwave Network Analyzer N9916A. The measured reflection coefficients (S11–S44) at ports #1, #2, #3, and #4 are displayed in Figure 14. The observed impedance bandwidth from the 4 to 6.5 GHz frequency band is 50% for a reflection coefficient less than −10 dB. The measured and simulated bandwidths of the multibeam antenna array are larger than the bandwidth of the antenna element because wideband feeding networks are present.

### 3.5. Measured Radiation Pattern

The anechoic chamber setup used to measure the far-field radiation pattern is shown in Figure 15. The VNA E8362B is connected to the AUT and the receiving horn antenna. A position controller was used to control the rotation of the AUT and the horn antenna, and a computer running a program the TY1200 software (version 2) was used to collect the measurement data. The measured 3-D radiation patterns are displayed in Figure 16. The excited port number, the direction of the beam, and the measured gain are used to identify each radiation pattern. At 5.2 GHz, the measured gain ranges from 9.1 dBi to 9.9 dBi.

The calculated and measured beam-pointing angles and gains for each excited port are listed in Table 3. When ports #1, #2, #3, and #4 are excited at 5.2 GHz, the observed beam-pointing angles are (30°, 335°), (20°, 50°), (20°, 205°), and (20°, 160°), respectively. Due to manufacturing tolerance and additional losses from the feeding network, the measured aiming angles (±30°, ±45°) deviate significantly from the predicted ones.

The designed multibeam array antenna’s simulated and measured 2-D radiation patterns are illustrated in Figure 17 for easier comparison. The red line shows the measured co-pol, whereas the blue line shows the simulated co-pol. The simulations and the measured results had a respectable level of agreement. The measured results support the finding that when port 1 is excited, the higher level of the second side lobe (SSL) is −9 dB.

Black is used to symbolize the simulated x-pol. The simulated x-pol at 5.2 GHz is less than −25 dB as ports #1 through #4 are excited.

### 3.6. Gain and Comparison

Figure 18 shows the multibeam array antenna’s measured and simulated gain. The measured gain is in the range of 6.5 to 9.8 dBi when port #1 is excited, whereas the simulated gain grows from 7.0 to 10.2 dBi over the frequency range of 3 to 7 GHz. Comparing the observed gain to the simulation, the gain decreased by 0.4 dBi. The uncertainty of the dielectric loss of the laminates used in manufacture is the main reason for the minor discrepancy between the measured and simulated gain. The connector losses are also uncalibrated.

The proposed 2 × 2 array module’s performance is compared to the existing reported 2-D multibeam array antennas that are realized using various guiding techniques. The comparison is summarized in Table 4. With low loss from the Panasonic Megtron 6 substrate and wideband from the BCS structure, the proposed 2-D is realized using broadside coupled stripline (BCS) technology, which exhibits good characteristics in the sub-6 GHz frequency spectrum. This feature results in a wideband impedance bandwidth of the array antenna larger than 50% at 5.2 GHz. In contrast, other designs implemented in SIW and PRGW technologies have lower impedance bandwidths than 40%, as indicated in Table 4 [11,17,29,34,35,36,37]. Moreover, the BCS allows for a compact BFN of size 0.25 λ_0_ × 0.25 λ_0_ as well as a compact array antenna of size 0.8 λ_0_ × 0.8 λ_0_, which is the most efficient design in terms of size, as shown in Table 4. The proposed multibeam antenna array has many noticeable merits, including a compact size, low loss, and superior impedance bandwidth performance.

## 4. Conclusions

This paper has proposed the design and measurement of a 2-D 4-beam array antenna fed by a 4 × 4 butler matrix in a BCS without a crossover and phase shifter at sub-6 GHz frequencies. The 2-D BFN was composed of just four broadside couplers. The expected amplitude distributions and phase responses in the frequency range of 3 to 7 GHz were verified by simulations and measured results. The 2-D BFN and 2 × 2 patch-array antenna demonstrated their ability to scan a beam by producing four orthogonal beams. A maximum simulated gain of 10.6 dBi and a measured gain of 9.6 dBi were achieved in (24°, 231°) and (20°, 205°) directions, respectively, when port 3 was excited. The 2-D BFN showed a compact physical size of 0.25 λ_0_ × 0.25 λ_0_ and exhibited a wideband operating frequency band of 66.7%. The array antenna’s physical area was reduced to 0.8 λ_0_ × 0.8 λ_0_ × 0.04 λ_0_ by exhibiting 50% matching bandwidth. The comparison between the calculation, simulation, and measurement showed agreement in terms of the S-parameters, radiation patterns, and realized gains. The 2-D beamforming array antenna is one of the most compact and promising choices for the 5G mobile communication system, according to performance comparisons with other designs.

## Figures and Tables

**Figure 1 sensors-24-00714-f001:**
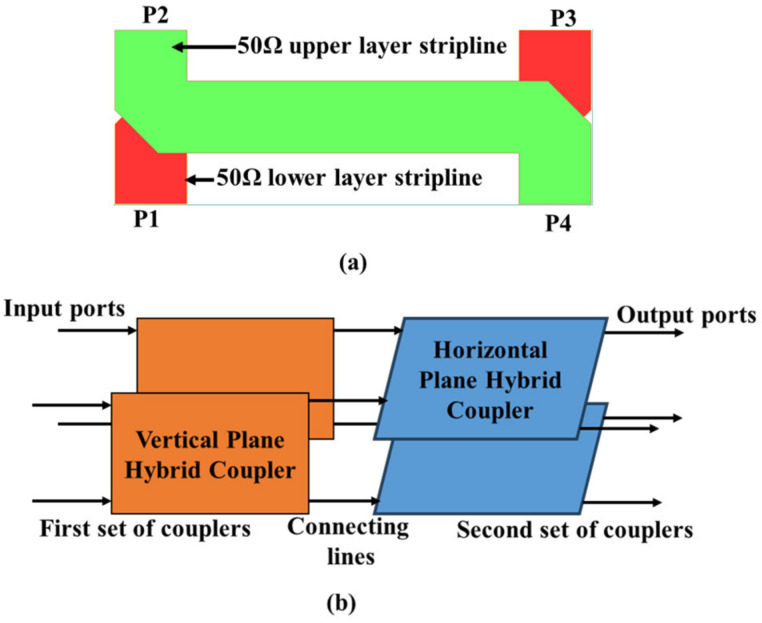
(**a**) Configuration of the vertical- and horizontal-plane couplers. (**b**) Topology of the 2-D BFN.

**Figure 2 sensors-24-00714-f002:**
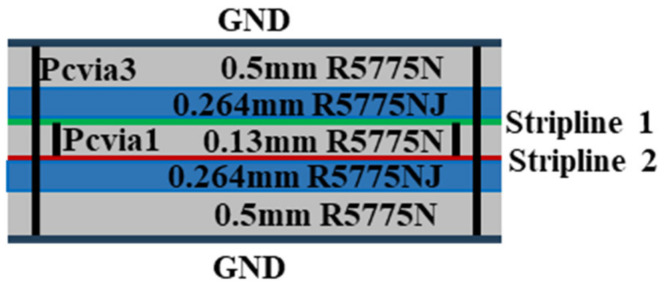
Cross-view of the proposed broadside coupled stripline.

**Figure 3 sensors-24-00714-f003:**
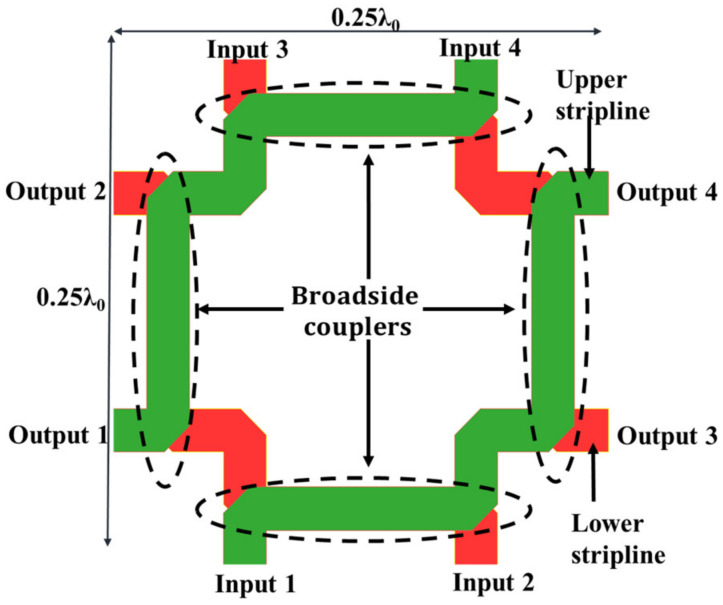
Proposed broadside coupled stripline 2-D BFN.

**Figure 4 sensors-24-00714-f004:**
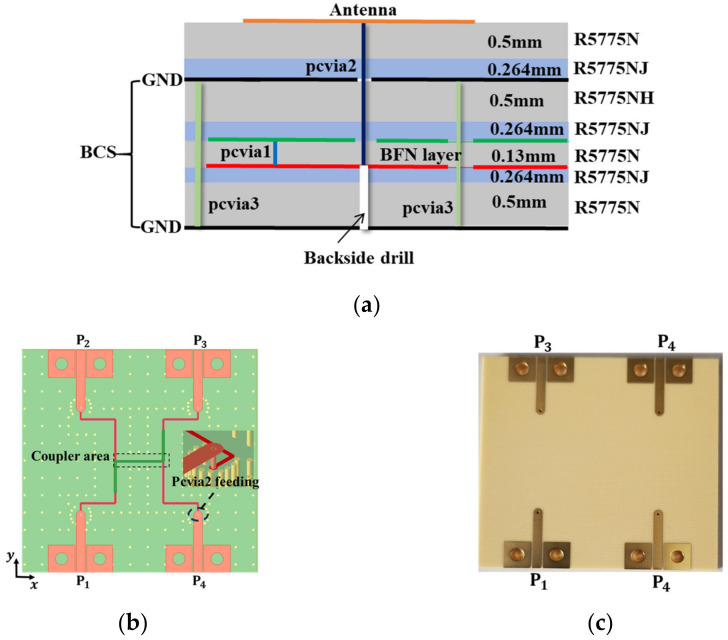
(**a**) Cross-view of the Panasonic Megtron 6 substrate; (**b**) coupler layout; and (**c**) fabricated prototype.

**Figure 5 sensors-24-00714-f005:**
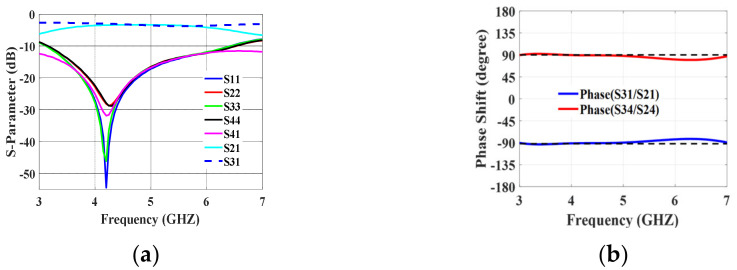
Broadside coupler simulation results: (**a**) S-paramters; (**b**) phase difference.

**Figure 6 sensors-24-00714-f006:**
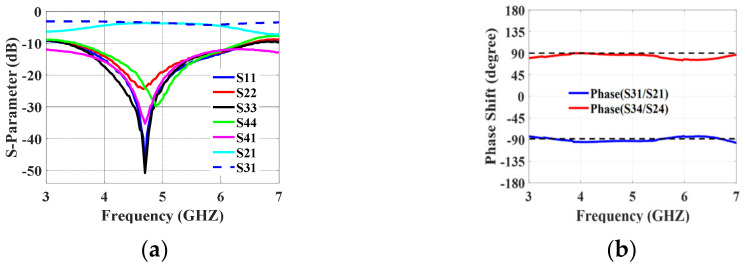
Broadside coupler measured results: (**a**) S-paramters; (**b**) phase difference.

**Figure 7 sensors-24-00714-f007:**
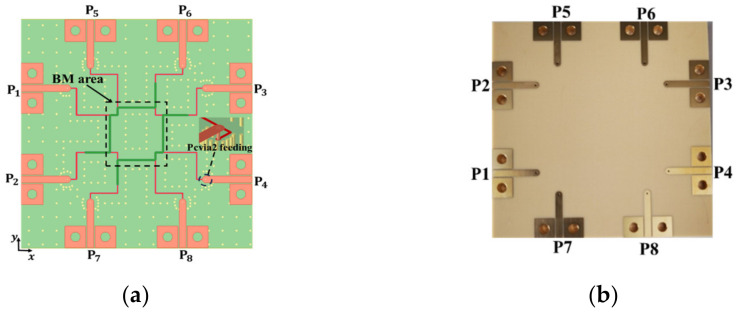
Proposed 2-D BFN: (**a**) simulation layout; (**b**) fabricated prototype.

**Figure 8 sensors-24-00714-f008:**
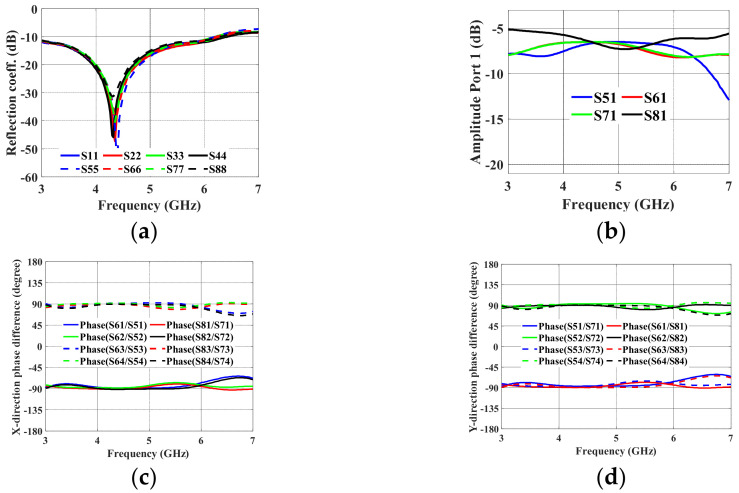
Proposed 2-D BFN simulation results: (**a**) reflection coefficient; (**b**) amplitude distribution; (**c**) phase shift in the *x*-direction; and (**d**) phase shift in the *y*-direction.

**Figure 9 sensors-24-00714-f009:**
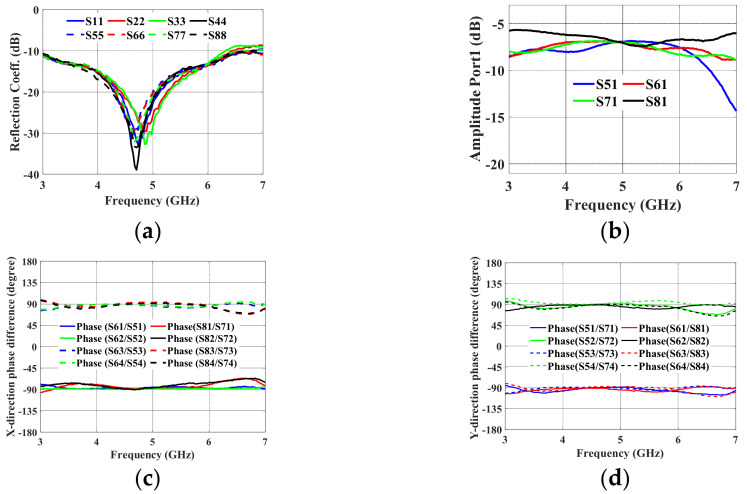
Proposed 2-D BFN measured results: (**a**) reflection coefficient; (**b**) amplitude distribution; (**c**) phase shift in the *x*-direction; and (**d**) phase shift in the *y*-direction.

**Figure 10 sensors-24-00714-f010:**
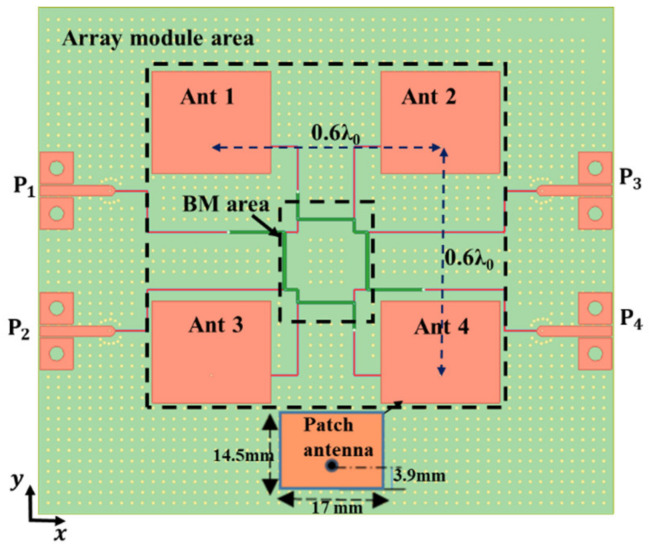
2 × 2 array design layout.

**Figure 11 sensors-24-00714-f011:**
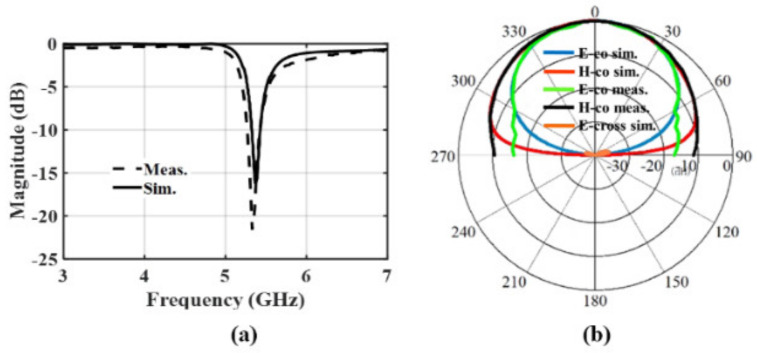
Antenna element: (**a**) simulated/measured reflection coefficients; (**b**) simulated E-(blue), H-(red), and E-cross (orange) planes and measured E-(green) and H-(black) planes at 5.2 GHz.

**Figure 12 sensors-24-00714-f012:**
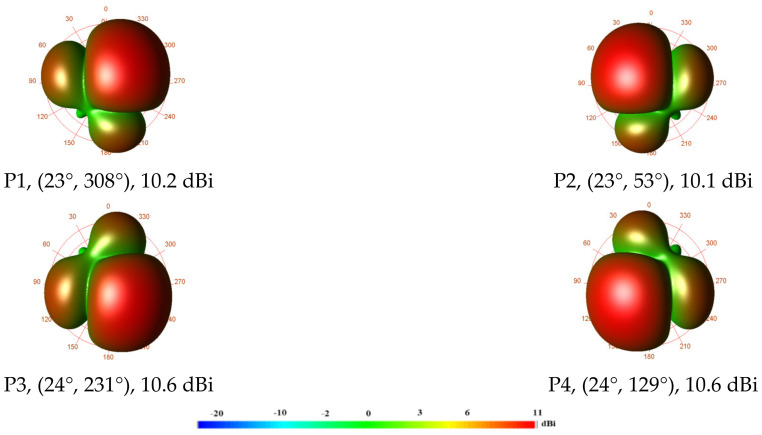
Simulated 3-D radiation patterns at 5.2 GHz.

**Figure 13 sensors-24-00714-f013:**
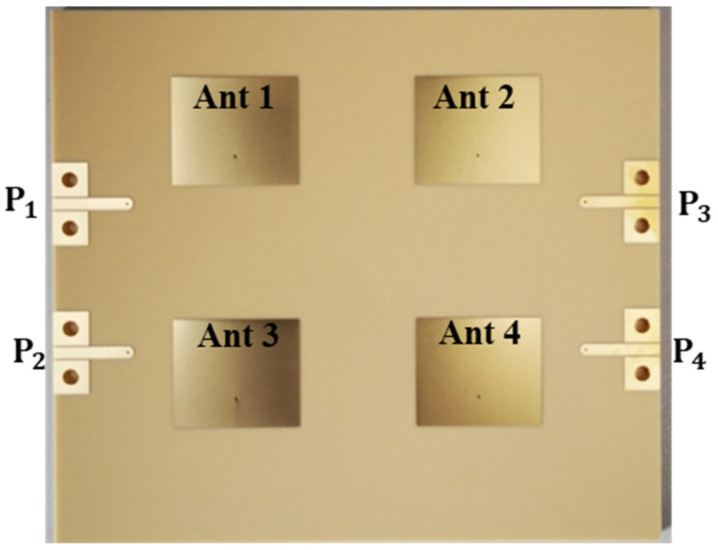
Fabricated prototype photography of the 2 × 2 array antenna.

**Figure 14 sensors-24-00714-f014:**
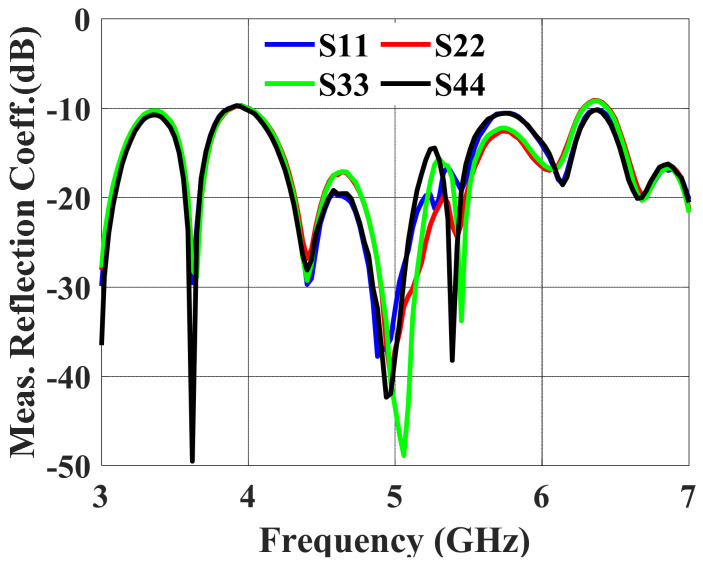
Measured reflection coefficients for ports #1, #2, #3, and #4.

**Figure 15 sensors-24-00714-f015:**
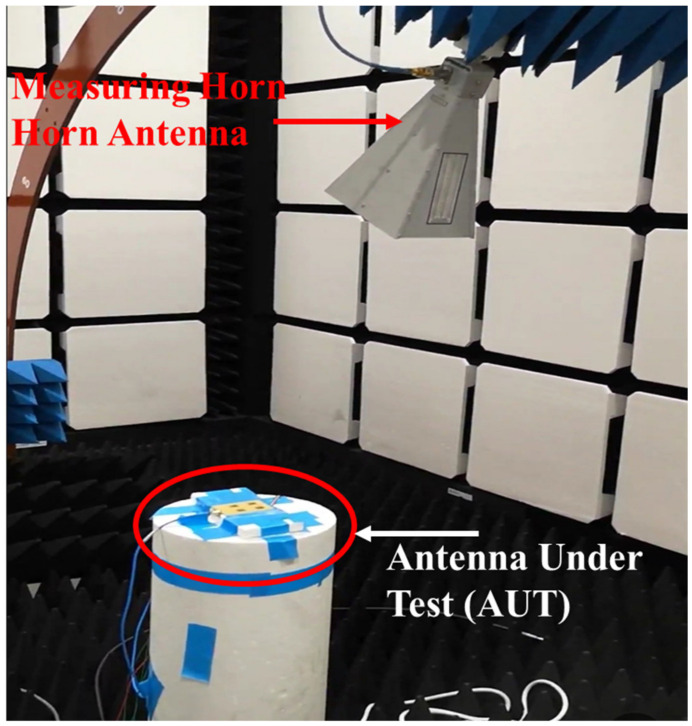
Anechoic chamber setup for radiation pattern measurement.

**Figure 16 sensors-24-00714-f016:**
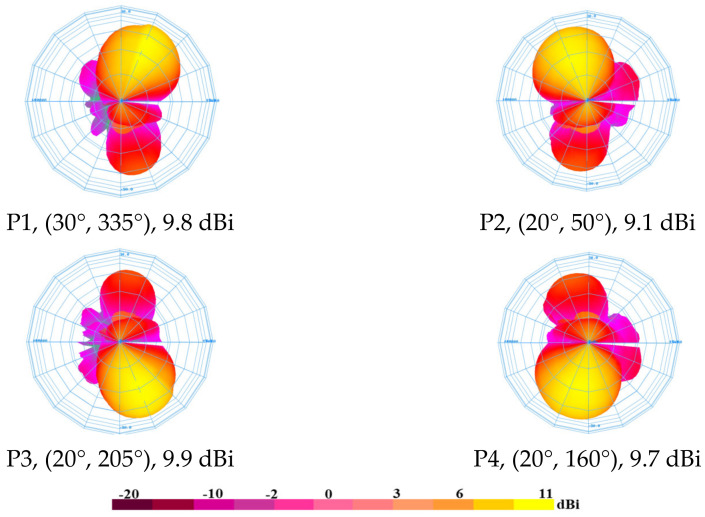
Measured 3-D radiation pattern at 5.2 GHz.

**Figure 17 sensors-24-00714-f017:**
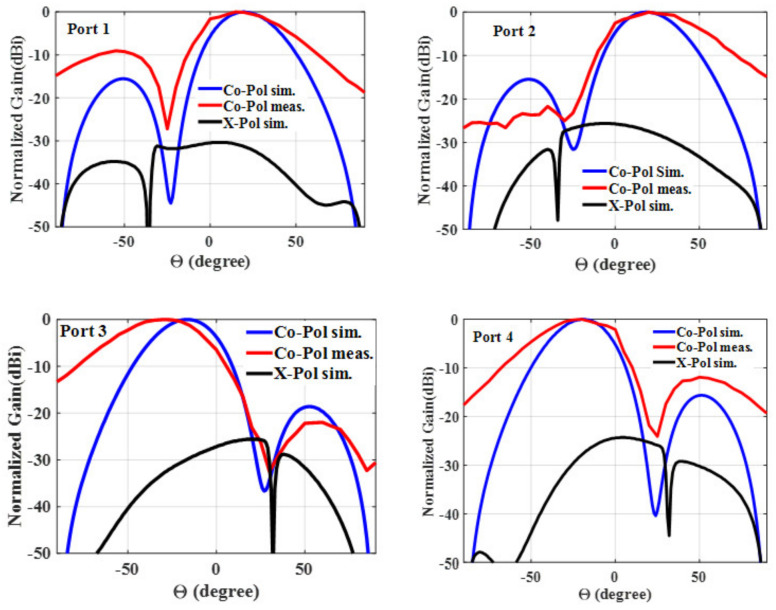
Simulated co-pol (blue), measured co-pol (red), and simulated x-pol (black) for ports #1, #2, #3, and #4 at 5.2 GHz.

**Figure 18 sensors-24-00714-f018:**
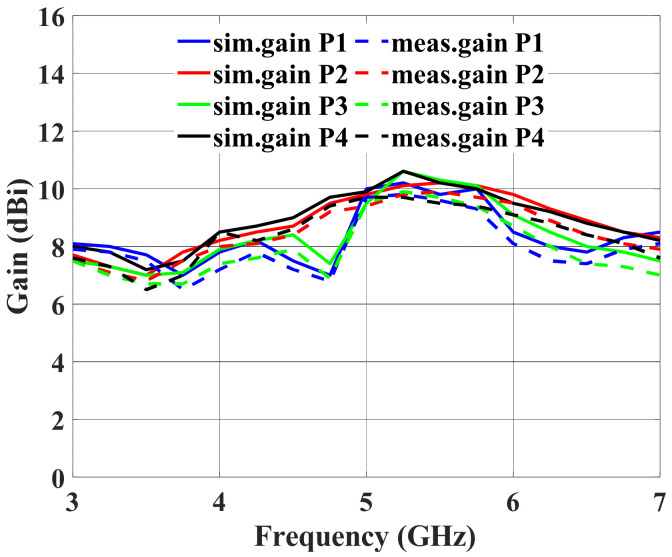
Simulated gain and measured gain when ports #1 to #4 are excited.

**Table 1 sensors-24-00714-t001:** Coupler theoretical phase shift in the *x*- and *y*-directions.

	P1	P2
$\alpha_{x}$	−90°	−90°
$\alpha_{y}$	−90°	90°

**Table 2 sensors-24-00714-t002:** Calculated and simulated beam directions.

Ports	$\left(\theta_{0c},\phi_{0c}\right)$	$\left(\theta_{0s},\phi_{0s}\right)$
P1	(30°, 315°)	(23°, 308°)
P2	(30°, 45°)	(23°, 53°)
P3	(30°, 225°)	(24°, 231°)
P4	(30°, 135°)	(24°, 129°)

**Table 3 sensors-24-00714-t003:** Comparison of simulated and measured beam directions.

	Simulation		Measurement	
Port	$(\theta_{0}$ $,\phi_{0})$	Gain	$(\theta_{0}$ $,\phi_{0})$	Gain
1	(23°, 308°)	10.2 dBi	(30°, 335°)	9.8 dBi
2	(23°, 54°)	10.1 dBi	(20°, 50°)	9.1 dBi
3	(24°, 231°)	10.6 dBi	(20°, 205°)	9.9 dBi
4	(24°, 129°)	10.6 dBi	(20°, 160°)	9.7 dBi

**Table 4 sensors-24-00714-t004:** Comparison with similar 2 × 2 array antennas.

Ref.	Guiding Tech.	Freq. (GHz)	BFN Size	Beams	Array Bandwidth	Module Size	Gain (dBi)
[11]	2-layer prgw	30	5.6 λ_0_ × 7.1 λ_0_	2 × 2 (ME-dipole)	20%	5.6 λ_0_ × 7.1λ_0_	10.3
[17]	4-layer SIW	26	2.9 λ_0_ × 2.9 λ_0_	2 × 2 (ring)	7.5%	2.9 λ_0_ × 2.9 λ_0_	12
[29]	3-layer Rectangular coaxial	30	6 λ_0_ × 6 λ_0_	2 × 2 (waveguide)	25.8%	6 λ_0_ × 6 λ_0_	16.8
[34]	2-layer SIW	28	Not indicated	2 × 2 (Diff-fed cavity-backed shorted patch)	14.3%	Not indicated	10.5
[35]	4-layer SIW	60	5.1 λ_0_ × 4.2 λ_0_	2 × 2 (cavity-backed patch)	34.5%	5.1 λ_0_ × 4.2 λ_0_	12.4
[36]	3-layer PCB	6	0.54 λ_0_ × 0.54 λ_0_	2 × 2 (E-shaped patch)	3.82%	1 λ_0_ × 1 λ_0_	11
[37]	1-layer PCB	5.5	2.97 λ_0_ × 2.97 λ_0_	2 × 2 (cp-fed circularly polarised square slot)	39%	2.97 λ_0_ × 2.97 λ_0_	10.1
This work	BCS	5.2	0.25 λ_0_ × 0.25 λ_0_	2 × 2 (Probe-fed patch)	50%	0.8 λ_0_ × 0.8 λ_0_	10.6

## Data Availability

Data are contained within the article.

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
