# Peer review of "A Compact Broadside Coupled Stripline 2-D Beamforming Network and Its Application to a 2-D Beam Scanning Array Antenna Using Panasonic Megtron 6 Substrate"

_sensors, 2024, doi:10.3390/s24020714_

Round 1

Reviewer 1 Report

Comments and Suggestions for Authors

1. Paper is well written.

2. In Fig.4 (b) & 7 (b) provide the rear view that will clarity of feeding into the system

3. How is the cross-polarisation purity of the antenna (measured)?

4. Fig. 17 provide polar plot for clarity.

5. Conclusion should be re-written with more clarity.

Author Response

The answers to the reviewer 1 questions are contained in the attached file.

Reviewer 2 Report

Comments and Suggestions for Authors

In this manuscript, authors designed a 2-D Butler Matrix for beamforming network. Both simulation and experiment results are in very good agreement. These works are significant in the 5th Generation of mobile communication system. According to this reviewer’s opinion, the manuscript can be accepted for publication unless the following changes are made. Figure 1 is apparently not professional enough. The reference data in Table 4 is not comprehensive enough and very old literature was selected and referenced.

Author Response

Answers to reviewer 2 is contained in the attached word file.

Reviewer 3 Report

Comments and Suggestions for Authors

1. Please revise Figures 1 and 2. It is recommended to draw original figures. The readability of the figures are low after printing.

2. Similar comment for Figures 3 and 4.

3. It is suggested to add the phase deviations from the absolute value to Figures 5 (b) and 6 (b) and reflecting consistent data to the body of the manuscript. If the authors believe that the phase shift at the desired frequency is the only important value, then please add a vertical line to the figure and write the value at the target frequency. Same comment for all the figures.

4. Please improve the quality of Figure 15.

5. In Table 4, the guiding technology of the work is different with the other references and makes the comparison difficult. It is suggested to add a similar reference to the table or describe the disadvantages and advantages of the different technologies. 

6. Please add more numerical performance parameters to the conclusion and emphasis the contribution of the work more clearly.

Author Response

Answers to reviewer 3 is contained in the attached word file.

Reviewer 4 Report

Comments and Suggestions for Authors

Journal Sensors (ISSN 1424-8220)

Manuscript ID sensors-2681123

Type Article

Title A Compact Broadside Coupled Stripline 2-D Beamforming Network And Its Application to 2-D Beam Scanning Array Antenna Using Panansonic Megtron 6 Substrate

Authors Jean Temga * , Shiba Takashi , Noriharu Suematsu

This paper addresses a 4-way 2-D Butler Matrix (BM)-based beamforming network (BFN) using multilayer substrate Broadside Coupled Stripline (BCS). The proposed BFN exhibits wide operating band of 66.7% (3-7 GHz) and a compact physical area of just 0.25λ0×0.25λ0. The manuscript is easy to read, and I can follow the authors points. I have the following comments for the authors.

1. The line spacing of the paper is not unified. Sometimes we have single line spacing, sometimes we have 1.5 lines spacing. Please revise.

2. Please indicate what is the working frequency range of your proposed BFN.

3. Can you explain Fig.14 a bit more? Why do we have so many low reflection coefficients in this frequency range?

4. In Fig.18, how do we understand the obtained results? Is there a large difference between the simulated gain and the actual gain?

5. A more detailed explanation about Table 4 is recommended.

 Author Response

Answers to reviewer 4 is contained in the attached word file.

Round 2

Reviewer 2 Report

Comments and Suggestions for Authors

In this revised manuscript, the authors have carefully revised the comments made and added useful information. it is very usful to the 2-D beamforming network and 2-d beam scanning array antenna using panasonic megtron 6 substrate. Essay writing meets specifications. This reviewer recommends for acceptance for publication.

Reviewer 3 Report

Comments and Suggestions for Authors

The authors have provided the requested revisions.